# Determination of Ploidy Levels and Nuclear DNA Content in *Cryptococcus neoformans* by Flow Cytometry: Drawbacks with Variability

**DOI:** 10.3390/jof10040296

**Published:** 2024-04-19

**Authors:** Yun C. Chang, Michael J. Davis, Kyung J. Kwon-Chung

**Affiliations:** Molecular Microbiology Section, Laboratory of Clinical Immunology and Microbiology, National Institute of Allergy and Infectious Diseases, National Institutes of Health, Bethesda, MD 20892, USA; ychang@nih.gov (Y.C.C.);

**Keywords:** ploidy determination, flow cytometry, propidium iodide, DAPI, SYTOX Green, SYBR Green I, imaging flowcytometry, Hoechst 33342

## Abstract

Flow cytometry is commonly employed for ploidy determination and cell cycle analysis in cryptococci. The cells are subjected to fixation and staining with DNA-binding fluorescent dyes, most commonly with propidium iodide (PI), before undergoing flow cytometric analysis. In ploidy determination, cell populations are classified according to variations in DNA content, as evidenced by the fluorescence intensity of stained cells. As reported in *Saccharomyces cerevisiae*, we found drawbacks with PI staining that confounded the accurate analysis of ploidy by flow cytometry when the size of the cryptococci changed significantly. However, the shift in the fluorescence intensity, unrelated to ploidy changes in cells with increased size, could be accurately interpreted by applying the ImageStream system. SYTOX Green or SYBR Green I, reported to enable DNA analysis with a higher accuracy than PI in *S. cerevisiae*, were nonspecific for nuclear DNA staining in cryptococci. Until dyes or methods capable of reducing the variability inherent in the drastic changes in cell size or shape become available, PI appears to remain the most reliable method for cell cycle or ploidy analysis in *Cryptococcus.*

## 1. Introduction

Isolates of *Cryptococcus neoformans* and *C. gattii* species complex typically exist as haploid yeasts with 14 chromosomes. However, both species complexes are recognized for their genome plasticity and frequent manifestation of variation in ploidy [1,2,3,4]. Aneuploidy is particularly pronounced when yeasts adapt to severe environmental stress in vitro and in vivo [5,6]. In addition, isolates resulting from mating between strains of genetically divergent lineages, such as those of molecular type VN1 (serotype A) and VNIV (serotype D) within the *C. neoformans* species complex, display notable ploidy variation [4,7]. Furthermore, polyploid cells (>4N), known as titan cells or giant cells, can be formed in vitro [8] and in the lungs or brains of mice during infection [9,10,11]. Given the significance of determining ploidy levels in various molecular genetic research projects and the impracticality of directly counting chromosomes in yeast cells, flow cytometry has become the standard method for assessing DNA ploidy in *Cryptococcus* species.

Flow cytometry assesses the cellular DNA content of cells by staining with fluorochromes such as propidium iodide (PI), 4′,6-diamidino-2-phenylindole (DAPI), or other DNA-specific dyes. These dyes bind stoichiometrically to nucleic acids, and the staining intensity is proportionate to the total DNA content [12,13,14]. PI and DAPI are commonly employed in cryptococcal research [4,7,9,15,16,17]. PI intercalates nonspecifically between bases in double-stranded nucleic acids, emitting red fluorescence when excited by blue light [13,14]. As it also binds to RNA, the enzymatic digestion of RNA is necessary to assess nuclear ploidy during fluorescence measurement [13,14]. In contrast, DAPI is more DNA-specific, binding nonintercalatively to repetitive AT-rich regions [18,19]. While useful for ploidy determination and cell cycle analysis, DAPI may not yield absolute DNA values or be suitable for comparing cells with varying AT base proportions [20].

Although PI remains the most widely used fluorochrome for DNA ploidy determination, its fluorescence intensity in stained samples was reported to be influenced by cell size or shape in *Saccharomyces cerevisiae* [21]. When cryptococci are exposed to fluconazole (FLC), there is a noticeable increase in cell size and changes in fluorescence intensity in flow cytometry analysis [22]. In this study, we analyzed the ploidy status of cryptococcal yeast cells exposed to fluconazole using propidium iodide (PI) and observed alterations in fluorescence intensity unrelated to ploidy. In addition, we explored the use of other fluorochromes, SYTOX Green [21] and SYBR Green I [23], which have been recommended for more accurate DNA analysis in *S. cerevisiae* [21]. We found that both SYTOX Green and SYBR Green were not useful for determining ploidy or analyzing the cell cycle in cryptococci, as their binding is nonspecific to nuclear DNA.

## 2. Methods

**Strains and culture conditions.** *Cryptococcus neoformans* strain H99 was grown in YPD broth to log phase at 30 °C. The log-phase cells were exposed to 32 μg/mL FLC for the indicated time to compare with naïve cells and were processed for staining with various fluorescent dyes.

**Propidium iodide staining**. PI staining was carried out according to established methods [22]. In short, approximately 2 × 10^7^ cells (1 OD_600_) were fixed in 70% ethanol overnight at 4 °C, collected, and suspended in NS buffer [24]. Subsequently, the sample was treated with RNase and stained with PI (Invitrogen, Carlsbad, CA, USA) at 37 °C for 2 h. The PI-stained cells were analyzed by flow cytometry as previously detailed [22]. ImageStream^®^ imaging flow cytometry analysis (Amnis Corporation, Seattle, WA, USA) was performed on H99 cells either untreated or treated with 32 µg/mL FLC for 8 h and stained with PI. Imperfect acquisition events (poor alignment, focus) as well as cell clumps were eliminated by gating in post-acquisition analysis using the ImageStream^®^ Ideas software version 4. To compare the most similar populations possible, we also gated only single (non-budding) cells.

**SYTOX Green and SYBR Green I staining.** The staining procedures for SYTOX Green and SYBR Green I (Invitrogen, Carlsbad, CA, USA) followed the methods established in the early 2000s [21,23]. Briefly, an approximate number of 2 × 10^7^ cells were fixed in 70% ethanol. For SYTOX Green staining, fixed cells were washed with H_2_O and resuspended in RNase solution, boiled for 15 min, allowed to cool at room temperature and incubated for 2–12 h at 37 °C. The cells were collected, resuspended in protease solution, and incubated for 15–20 min. Protease-treated cells were then added to SYTOX Green solution, sonicated, and subjected to analysis by standard flow cytometry methods [21]. For SYBR Green I staining, fixed and washed cells were incubated in RNase solution for 1 h at 50 °C. Proteinase treatment was also performed at 50 °C for 1 h before staining with SYBR Green I at 6 °C to 8 °C overnight in the dark to protect from light. Stained cells were sonicated and subjected to flow cytometry analysis [23].

In addition, ethanol-fixed cells were stained with 3 μg/mL DAPI and subsequently were subjected to flow cytometry analysis. For Hoechst 33258 or Hoechst 33342 (Invitrogen, Carlsbad, CA, USA) staining, a final concentration of 10 μg/mL of the dye was added to the cultures of live cells for 30 min and then were subjected to flow cytometry analysis.

## 3. Results and Discussions

We first examined PI-stained *C. neoformans* cells which had been exposed to fluconazole (FLC). The FLC-untreated sample displays the typical G1 and G2 peaks (Figure 1A, blue color), while the sample treated with FLC displays increased fluorescent intensity (Figure 1A, red color). The increased fluorescence intensity in the FLC-treated sample correlated with the increased cell size inferred from the forward scatter plots (Figure 1B–D). These results were similar to the previous observation that there is ambiguity in the flow cytometric determination of the DNA content in fluconazole-treated *C. neoformans* cells stained with propidium iodide [22]. However, ImageStream imaging flow cytometry analysis after applying the nuclear mask method revealed that the presumed polyploidy status in the FLC-treated samples was primarily attributed to an elevated cytoplasmic PI signal unrelated to ploidy (Figure 2).

The ImageStream system is a sophisticated flow cytometer capable of capturing fluorescence signals and high-quality fluorescence images, facilitating multiparameter analysis [25]. Analyzing images with ImageStream necessitates creating masks that divide the images into areas of interest (events/cells) and background. By default, the software generates masks encompassing the entire cell. This default mask is similar to the signals produced by flow cytometry.

Figure 2A demonstrates that the alteration in the fluorescence intensity pattern upon the FLC treatment in the ImageStream analysis, when employing the software’s default mask, closely resembles the pattern observed in conventional flow cytometry (Figure 1A). The “>2N” gate in the FLC-treated condition is displayed to indicate that these cells appear to have a DNA content exceeding 2N, potentially indicative of “polyploidy”. We then used the spot function in the Ideas software to create a new mask to isolate each cell’s nucleus from the surrounding cytoplasm. A further mask isolating each cell’s cytoplasm was created by subtracting the nuclear mask from the whole-cell mask. Figure 2B shows the sample images of a typical cell displaying the brightfield image (i), default software whole cell (ii), nuclear mask (iii), and cytoplasmic mask (iv). The apparent putative >2N population in panel A disappears after applying the nuclear mask (Figure 2C), suggesting that the increased fluorescent intensity in the putative “>2N” area is in fact not due to a ploidy change.

Two representative cells from the FLC-treated sample (cell x from near the G2 peak and cell y from the “>2N” area in Figure 2D left panel) were selected to further illustrate the effects of generating a mask using different methods in the Ideas software. The fluorescence intensities of example cells x and y using the whole-cell, nuclear, and cytoplasmic masks are displayed in the right panel of Figure 2E. It is evident that the cytoplasmic fluorescence intensity is higher in cell y than in cell x, which contributes to the major fluorescent intensity increase observed in the whole cell. The dot plot analysis further demonstrates that the cells from the >2N gate have a larger area of cytoplasm as well as cytoplasmic fluorescent intensity but do not display a clear increase in the actual nuclear fluorescence intensity (Figure 2F,G). To sum up, the cells in the >2N gate which could be interpreted as containing higher ploidy are mainly the cells with increased cytoplasmic fluorescent intensity. These findings suggest that determining ploidy in PI-stained cryptococcal cells displaying changes in size or shape cannot solely rely on simple flow cytometry data.

In work with *S. cerevisiae*, PI-stained cells commonly exhibit increases in the coefficients of variations in fluorescence associated with an increase in cell size or changes in morphology [21]. In our study, an imaging flow cytometry proved to be valuable in distinguishing the fluorescent signals of PI in the cytoplasm versus the nucleus. PI-associated fluorescence shifts in flow cytometry, unrelated to ploidy changes, have also been documented in cryptococcal strains with specific gene mutations leading to alterations in cell size [25,26]. This observation supports that an alteration in the cryptococcal cell size can distort the ploidy status and emphasize the importance of examining cell morphology prior to flow cytometry. The microscopic examination of cells would suffice to detect changes in the size or shape of the cells. If microscopic examination was omitted, drastic alterations in scatter values in flow cytometry may indicate changes in the cell size or shape. In such cases, supporting cytological, genetic, or other evidence is essential before interpreting a fluorescence shift as indicative of an alteration in ploidy.

To address the variability associated with PI staining in the FL-treated cells, we explored alternative dyes, namely, SYTOX Green and SYBR Green I. These dyes are known to bind to DNA/RNA without base specificity [13]. They are commonly used in bacterial staining for viability [27] and in studies of *S. cerevisiae*’s cell cycle and ploidy. SYTOX Green and SYBER Green 1 staining are considered superior for ploidy analysis in *S. cerevisiae* because they reduce the variability associated with PI staining [21,23]. Unlike the case of *S. cerevisiae*, however, the staining patterns of the cryptococcal cells with these dyes were inconsistent in that some cells were successfully stained while others failed to be stained. Furthermore, the dyes did not specifically label nuclear DNA in the RNase A-treated cells and yielded unreliable flow cytometric data (Figure 3 and unpublished observations). This was unexpected since SYTOX Green has a higher affinity to DNA than PI [21] at working concentrations in other systems, and the stain readily penetrates the compromised plasma membranes of alcohol-fixed cells. SYBR Green I [23] behaved similarly to SYTOX Green in terms of the inconsistency in the staining pattern and showed no specificity for the nuclear DNA of the cryptococcal cells (Figure 3). *S. cerevisiae* and *Cryptococcus* are phylogenetically distanced yeasts with differences in cell wall and membrane compositions. In addition, cryptococcal cells are surrounded by a thick polysaccharide capsule. We speculate that the permeation of the two dyes is not as efficient as in *S. cerevisiae* due to these differences and the results of inconsistent cell staining. The two dyes not being DNA-specific in stained cryptococci is an enigma at this point and requires further study.

Although DAPI, Hoechst 33342, and Hoechst 33258 have been used for ploidy determination [13,14], all three of these dyes showed inferior G1/G2 signal separation compared to PI (Figure 4). Moreover, the flow cytometric patterns of the DAPI-, Hoechst 33342-, and Hoechst 33258-stained cryptococcal cells were all influenced by growth conditions as an FLC treatment (Figure 4). These results indicated that the nuclear-DNA-binding dyes that can be reliably used for ploidy determination or cell cycle analysis in cryptococcal cells are more limited than for *S. cerevisiae*. Until the dyes or methods which can reduce the variability inherent in the drawbacks discussed above become available, PI appears to remain the most reliable method for cell cycle or ploidy analysis in *Cryptococcus*, provided that the isolates manifest no drastic changes in size or shape.

In conclusion, the selection of DNA-binding dyes suitable for cryptococcal ploidy determination or cell cycle analysis by flow cytometry is more limited compared to that for *S. cerevisiae*. Although PI staining seems to be the most viable method currently available for cryptococcal cell cycle or ploidy analysis, it is important to note that alterations in cell size or shape may lead to distortions in the interpretation of ploidy status.

## Figures and Tables

**Figure 1 jof-10-00296-f001:**
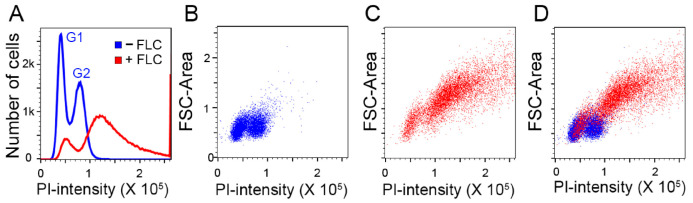
FLC treatment causes changes in the flow cytometry patterns in PI-stained cells. H99 cells grown to log phase were treated with 32 µg/mL of FLC for 8 h, stained with PI, and subjected to flow cytometry. Doublets were removed by gating in post-acquisition analysis. (**A**) Histograms of FLC-treated (red) and untreated control (blue). The FLC-untreated sample displays the typical G1 and G2 peaks. The sample treated with FLC displays increased fluorescent intensity. (**B**–**D**) Dot plots displaying PI fluorescence intensity (*x*-axis) and forward scatter area (*y*-axis). Untreated control (blue; **B**), FLC-treated sample (red; **C**), overlay of the two populations (**D**). The increased fluorescence intensity in FLC-treated sample is correlated with increased cell size inferred from forward scatter area (FSC-Area).

**Figure 2 jof-10-00296-f002:**
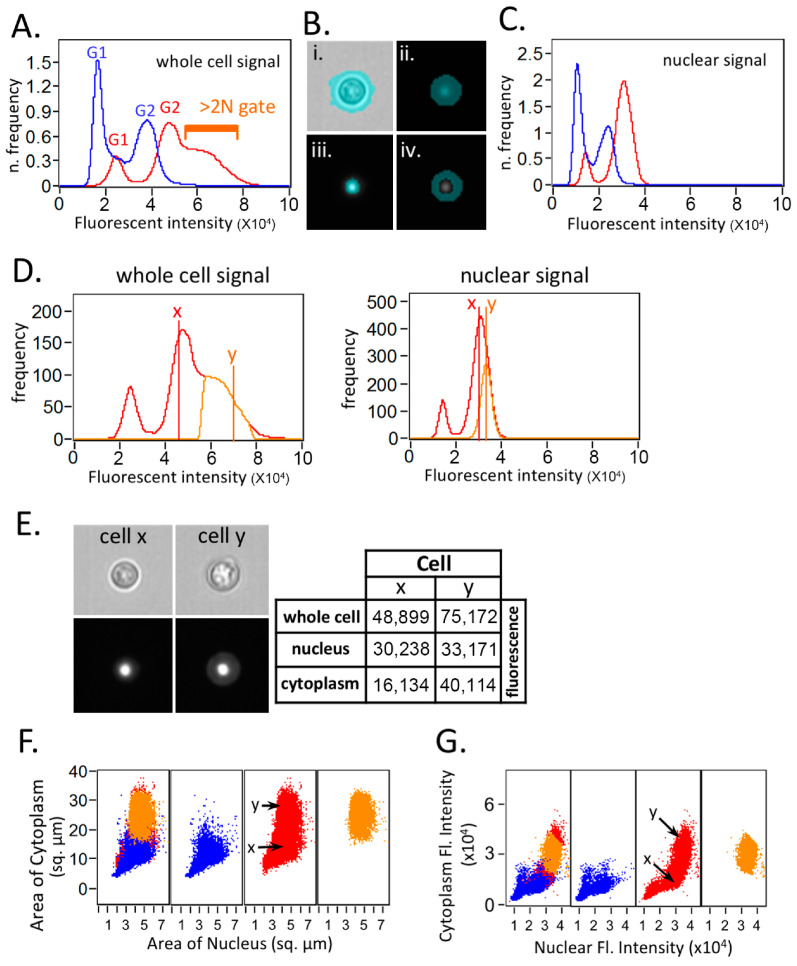
Imaging flow cytometry analysis suggests that most of the putative polyploidy events observed by flow cytometry in FLC-treated samples are due to increased cytoplasmic PI signal. ImageStream imaging flow cytometric analysis was performed on H99 cells either treated or untreated with 32 µg/mL FLC for 8 h and stained with PI. (**A**) Histograms of FLC-treated (red) and untreated control (blue) PI fluorescence intensity signal using software default mask. The “>2N” gate in the FLC-treated condition is displayed to indicate that these cells appear to have DNA content >2N, which might be putative “polypoid”. (**B**) New masks were created using “spot” function in the Ideas software to isolate each cell’s nucleus from the surrounding cytoplasm. Sample images of a typical cell displaying the brightfield image (**i**), default software whole cell (**ii**), nuclear (**iii**) and cytoplasmic masks (**iv**). (**C**) Histograms of FLC-treated (red) and -untreated control (blue) histograms using nuclear mask for the PI channel. The apparent putative >2N population in panel A disappears after applying the nuclear mask. (**D**) Histogram display of fluorescence intensity signal. Left panel: whole-cell fluorescence signal from the FLC-treated sample showing the “>2N” gated (orange) and the complete set of FLC-treated events (red). Position of example cells x and y are highlighted. Right panel: nuclear fluorescence signal from the FLC-treated sample showing the >2N gated (orange) and the complete set of FLC-treated events (red). Position of example cells x and y are highlighted. (**E**) Analysis of two representative cells from the FLC-treated sample. Cell x is from near the G2 peak and cell y is from the “>2N” area in 2D. Brightfield (upper left panel) and fluorescence images (lower left panel) of each cell. Fluorescence images are identically scaled for comparison. Table displaying the fluorescence intensities of example cells x and y using whole-cell, nuclear, and cytoplasmic masks, respectively. (**F**) Dot plots displaying area of cytoplasmic mask (*y*-axis) and area of nuclear mask (*x*-axis) in square micrometers (left panel). Panels display untreated control (blue; center left), FLC-treated (red; center right), >2N gated FLC-treated (orange; far right) and an overlay of the three populations (far left). Positions of the example cells “x” and “y” are marked. (**G**) Dot plots displaying the fluorescence intensity inside the cytoplasmic mask (*y*-axis) and the fluorescence intensity inside the nuclear mask (*x*-axis). Panels display untreated control (blue; center left), FLC-treated (red; center right), >2N gated FLC-treated (orange; far right), and an overlay of the three populations (far left). Positions of the example cells “x” and “y” are marked. It is clear that the cells from the >2N gate have a larger area of cytoplasm and cytoplasmic fluorescent intensity but do not have a clear increase in actual nuclear fluorescence intensity.

**Figure 3 jof-10-00296-f003:**
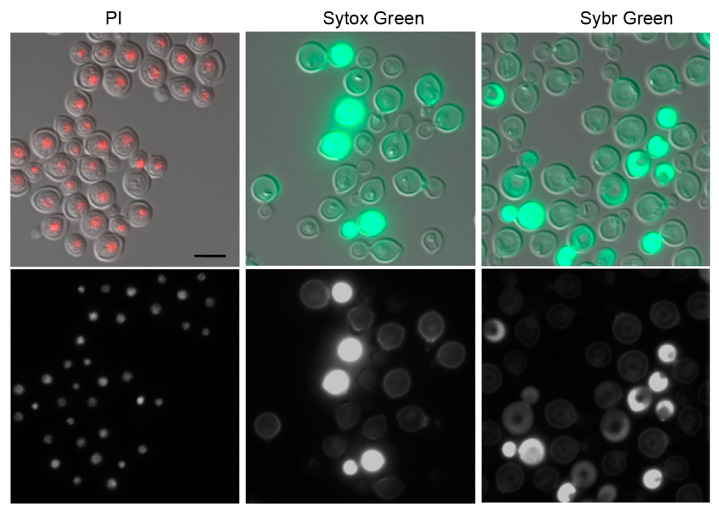
Micrographs of cells stained with different fluorescent dyes. H99 cells were grown to log phase, fixed in 70% ethanol, and stained with indicated dyes. PI-stained sample shows good signal-to-noise ratio. SYTOX Green- and SYBR Green I-stained samples show inconsistent cell staining and the nuclei are stained poorly. Upper panels: merged DIC and fluorescent images. Lower panel: fluorescent images. Bar = 5 μm.

**Figure 4 jof-10-00296-f004:**
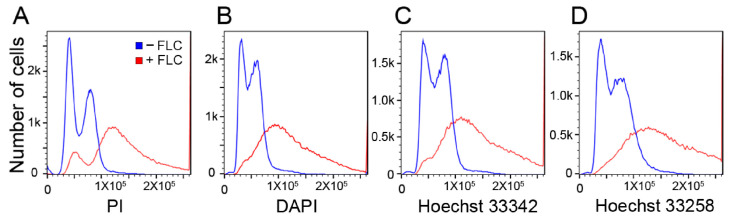
The flow cytometry patterns are similar in four different fluorescent dyes. H99 cells grown to log phase were treated with 32 µg/mL of FLC for 8 h, stained with various dyes, and subjected to flow cytometry. Histograms show FLC-treated (red) and -untreated control (blue) samples. Cells were stained with PI (**A**), DAPI (**B**), Hoechst 3342 (**C**), and Hoechst 33258 (**D**).

## Data Availability

Data are contained within the article.

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
