# Peer review of "Determination of Ploidy Levels and Nuclear DNA Content in Cryptococcus neoformans by Flow Cytometry: Drawbacks with Variability"

_jof, 2024, doi:10.3390/jof10040296_

Round 1

Reviewer 1 Report

Comments and Suggestions for Authors

Presented paper is a good example of brief communication regarding features and restrictions of a useful experimental procedure. This is exactly the part of science that needs to be disseminated and supported more. Such communications can save a lot of time for researchers and help them to avoid mistakes.

I have found only small mistakes that need correction:

36-37 and further - “in vitro and in vivo” usually is written in italic

63 -“use of another fluorochrome, SYTOX Green [21] and SYBR Green I” – maybe “fluorochromes”

69  – “The log phase ells

70 – “naïve

77-78 “flow cytometry analysis (Amnis Corporation, Seattle, WA) was performed on H99 cells  either treated or untreated with 32 µg/ml FLC for 8h and stained with PI.” – please rephrase, maybe “either untreated  or treated with 32 µg/ml FLC“

90-92 “Proteinase treatment was also at 50oC for 1h before staining with SYBR Green I at 6oC to 8oC overnight in dark to protect from light.” – please rephrase.

94 - “were stained with 3mg/ml DAPI or live cells were” – maybe “were stained with 3mg/ml DAPI and live cells were”

Reviewer 2 Report

Comments and Suggestions for Authors

This is a study comparing the appropriateness of different nucleic acid staining techniques-dyes to detect polyploidy in cryptococcus cells after treatment with fluconazole. However several points need to be further addressed.

A more detailed comparison of the different dyes regarding their DNA binding capacity in terms of eg compaction status needs to be provided, as well as the effect that this might effectuate on the procedure.

The dyes Hoechst are barely explained in the Methods section.

The clinical significance of the appropriateness of the detection method used for polyploidy needs to be elaborated upon.

The discrepancy of these results to the previously reported ones on Saccharomyces regarding the efficacy of SYTOX Green and SYBR Green I staining needs to be explained.

The manuscript needs minor language editing.

Comments on the Quality of English Language

Minor editing required

eg line 20 (abstract) should read confounded instead of confound

Round 2

Reviewer 2 Report

Comments and Suggestions for Authors

The authors have addressed the points adequately.